# Challenges and Adverse Effects of Wearing Face Masks in the COVID-19 Era

**Francis Gyapong** [1,2], **Ethel Debrah** [1,2], **Maame Oforiwaa** [1,2], **Abiola Isawumi** [1,2,*] **and Lydia Mosi** [1,2]

1   West African Centre for Cell Biology of Infectious Pathogens, College of Basic and Applied Sciences, University of Ghana, Legon, Accra P.O. Box LG 54, Ghana
2   Department of Biochemistry, Cell and Molecular Biology, College of Basic and Applied Sciences, University of Ghana, Legon, Accra P.O. Box LG 54, Ghana
*   Correspondence: isawumiabiola@gmail.com

**Abstract:** Background: The use of face masks was a significant part of the WHO COVID-19 preventive protocols. While their usage has been effective, lack of adherence by individuals has been associated with discomfort and adverse side effects. This might facilitate unnecessary exposure to the SARS-CoV-2 virus, thereby increasing the incidence of COVID-19. This study assessed the side effects of prolonged mask-wearing and offers recommendations for present and future pandemics. Methods: Adverse side effects of face masks were evaluated from November 2021 to February 2022 with a structured Google Forms online questionnaire. The survey targeted regular and occasional face mask users around the world. All responders anonymously completed the survey, which included ten structured questions with a sub-section on the effects of the continuous use of face masks. The information obtained was analyzed using descriptive statistics, and the data were presented in graphs. Results: Almost 60% (1243) of the 2136 participants indicated discomfort while using face masks. Breathing difficulties and pain around the ears were cited as major causes of discomfort, accounting for 32% and 22%, respectively, of responses. Headaches were reported by 26.8% (572) of the respondents, with 44.6% experiencing one within 1 h of wearing a mask. Nine hundred and eight (908) respondents experienced nasal discomfort, while 412 individuals reported various skin-related discomfort, including excessive sweating around the mouth and acne. Conclusions: This study provides baseline data as to why there was less adherence to face mask use which includes headaches, skin irritation, ear pains, breathing difficulties, sore throat, dry eyes, and increased sweating around the mouth. As a result, this may contribute to an increased risk of infection. While COVID-19 lingers and the management of its undesirable effects persists into the future, it is vital that a superior mask design, concentrating on safety, comfort, and tolerability, be developed.

**Keywords:** COVID-19; nose masks; breathing difficulties; side effects; public health; face masks





## 1. Introduction

Coronavirus disease 2019 (COVID-19) is the sixth public health emergency of global concern and is described as one of the most alarming pandemics of the 21st century [1,2]. Since its emergence in Wuhan, China, COVID-19, caused by a novel coronavirus, severe acute respiratory syndrome coronavirus 2 (SARS-CoV-2), has had 572,239,451 confirmed cases and has killed 6,390,401 globally [3] as of August 2022. As SARS-CoV-2 keeps evolving, Africa has recorded the lowest number of cases at 9,209,133 and a death rate of 173,974 as of August 2022. To contain and control the spread of COVID-19, the WHO instituted stringent preventive protocols, including wearing face/nose masks, frequent handwashing, and social distancing, among others [4].

The mandatory wearing of masks appears to be an efficient non-pharmaceutical intervention [5], as the primary transmission route of COVID-19 is via either droplets that infect the upper respiratory tract (with a short lifespan) or finer aerosols (which may persist

for hours) in the air [6]. This protocol has been described as effective [7]; however, wearing face masks has been associated with some physiological side effects which have prevented its efficient usage [4,8], thereby facilitating increased exposure to the SARS-CoV-2 virus. Moreover, prolonged face mask usage increases the temperature beneath the mask and hence becomes increasingly uncomfortable [9]. Previous studies on nose-mask-associated adverse effects have been reported among healthcare workers. These include headaches, vision obstruction, skin irritation and deterioration (pimples, itches, and rashes), facial pain and obstructions to vision, thermal equilibrium, and communication [9–12].

The increased $CO_2$ accumulation within masks has been associated with impaired cognition and disorientation [13]. This suggests low blood oxygen levels and could result in anaemia and acute respiratory distress syndrome upon prolonged mask use [14]. Moreover, formation of acne has been linked to microbiome dysbiosis resulting from increased humidity and bacteria entrapment in the mask space. Previous studies on face-mask-associated side effects have been reported among healthcare workers; however, there is a paucity of data from the community setting. This represents a public health concern as masks have become part of our social life. Therefore, this study profiled users' behaviors and frequent side effects preventing the effective use of face masks in the COVID-19 era.

## 2. Materials and Methods

### 2.1. Study Design and Data Collection Tools

An online structured questionnaire was developed using Google Forms targeting regular and occasional face mask users worldwide (Supplementary File S1-Questionnaire format). The questionnaire was shared electronically through various online platforms (including Facebook, Twitter, WhatsApp, Telegram, etc.). A total of 2136 respondents were assessed, and the data were obtained for analysis. All responders anonymously completed the survey, which included health risk informed questions on the effects of short and prolonged/continuous use of face masks. Physiological parameters including possible pain in the nose/nostrils (nasal irritation, dry/itchy nose, altered sense of smell), acne, skin changes (redness on the face), ear pains, dry mouth, bad breath, sore throat, breathing difficulties, dry eyes, tearing, and excessive sweating were assessed for data collection.

### 2.2. Statistical Analysis

The outcome of the data was assembled and statistically assessed using MS Excel 2021 LTSC (version 2206). Descriptive statistics were used in this study (with SPSS 16.0 and GraphPad 6.0), and the data were presented in graphs. The data were subjected to a chi-square test of proportions to estimate the normal distribution. The significance level was computed at $p < 0.05$.

## 3. Results and Discussion

A total of 2136 participants participated in this study, with 1243 (58.2%) indicating that nose mask usage makes them uncomfortable (Figure 1A). Breathing difficulties and pain around the ears were the most represented reasons for discomfort, accounting for 32% and 22%, respectively, of responses. Foggy glasses, frequent touching, and general nose mask safety issues were other reasons for discomfort (Figure 1B). Reported side effects included headaches, skin irritation/infections, ear pains, breathing difficulties, increased sweating, nasal discomfort, dry eyes, bad breath, and sore throat. The highest reported side effect was breathing difficulties, with 65.2% (1393) of the respondents reporting this adverse effect ($p < 0.001$) (Supplementary Table S1). About 63.4% (883) indicated that breathing difficulties occurred within 1 h of wearing a mask, 19.4% (270) indicated that they occurred within 1 to 3 h, and 17.2 % (240) indicated that they occurred after 3 h or more (Supplementary Table S1).

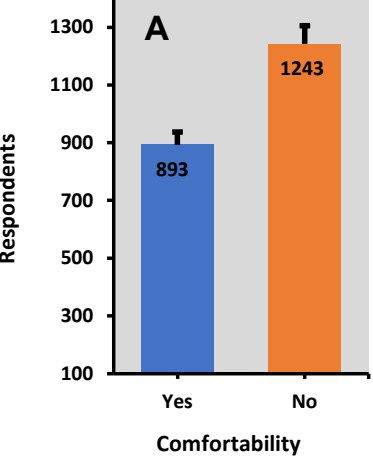
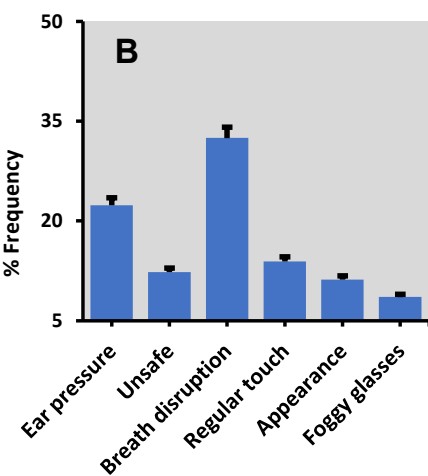

**Figure 1.** Nose mask usage as indicated by the respondents. (**A**) Comfortability and non-comfortability of nose masks. (**B**) Factors for the discomfort of nose mask usage. Error bars indicate the number of participants comfortable with nose masks (**A**) and the percentage of discomfort (**B**).

Most respondents, totaling 76.4% (1064), indicated that breathing resistance was mild, and 34% (743) reported no breathing resistance from regular mask use (Figure 2). About 72.3% (1564) reported no headaches from prolonged mask use. Of the participants, 26.8% (572) reported headaches: 60% (343) indicated their occurrence within 1 h of wearing a mask, 31.8% (182) indicated their occurrence within 1 to 3 h, and 8.2% (47) indicated their occurrence after 3 h or more (Supplementary Table S1). Out of the 2136 participants, 42.5% experienced nasal discomfort. The respondents reported one or more kinds of nasal discomfort. About 60% experienced generalized nasal discomfort, 43.4% developed an itchy nose, 36.8% experienced a burning sensation in the nose, and 23.7% developed a dry nose. Unusual nasal difficulties, such as blood after using tissues and an altered sense of smell, were reported by 21% of the participants. Some participants experienced pressure complications from prolonged mask use, including pain in the nose and ears (Figure 2 and Supplementary Table S1).

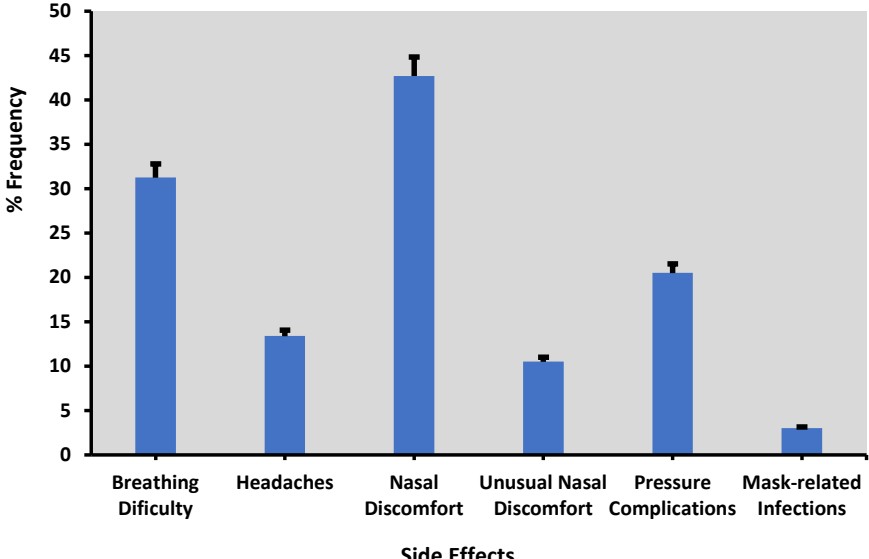

**Figure 2.** Side effects of nose mask usage as indicated by the respondents. Error bars represent the percentage of reported side effects.

Skin-related discomfort was reported by 19.3% of the participants, which includes excessive sweating around the mouth (41%), acne (23%), rashes (20%), and itchy skin (16%) (Figure 3A). Subsequently, 64% of the participants stated that these conditions resolved themselves independently. However, cold medicines and other ointments were used appropriately to treat conditions such as colds and rashes. Other side effects included bad breath (21%), nasal blockage (20%), dry mouth (14%), sore throat (9%), and dry eyes (7%) (Figure 3B). More than 90% of the participants who regularly used face masks have not had any case of COVID-19 infection. However, about 8.1% (173) reported COVID-19 infection irrespective of face mask usage (Figure 4A), with azithromycin (53%) as the most used antibiotic for treatment (Figure 4B).

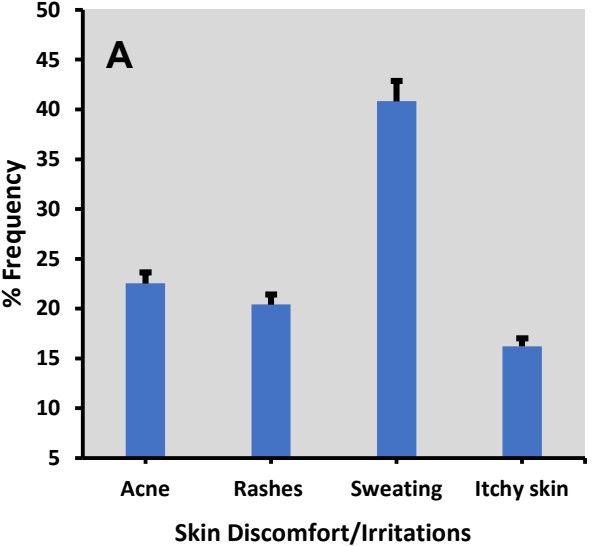
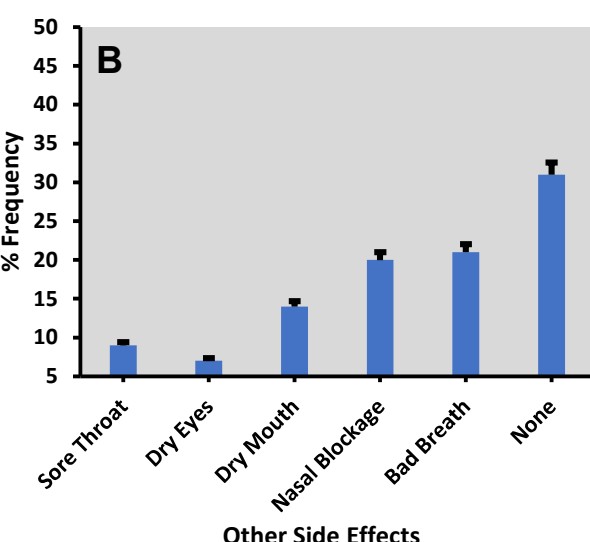

**Figure 3.** (**A**) Skin discomfort/irritation. (**B**) Other side effects as indicated by the respondents. Error bars represent the percentage of skin discomfort and other related side effects.

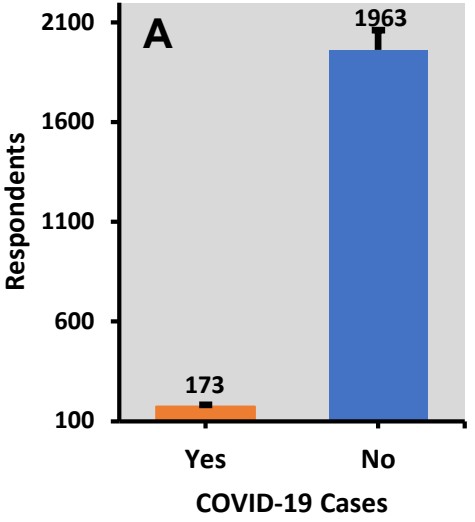
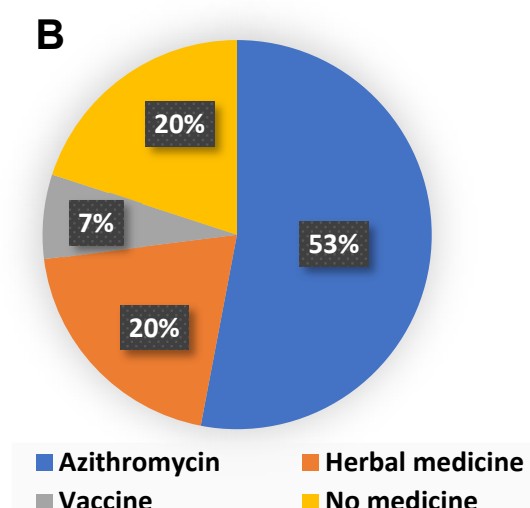

**Figure 4.** Incidence of COVID-19 cases as indicated by the respondents. (**A**) Reports of COVID-19 infection. (**B**) Treatments received for COVID-19 infection. Error bars represent the percentage of COVID-19 cases.

More than 40% of the participants indicated adverse reactions to prolonged mask use during the COVID-19 era. This included breathing difficulties, headaches, pressure complications, mask-related infections, skin irritation, and facial and nasal discomfort.

In this study, almost 60% of the participants had generalized nasal discomfort using a mask. This might be attributable to the hot and humid air beneath the mask's dead space compared to the ambient temperature. It has been established that a decrease in air humidity beneath the face mask and a reduction in the skin's transpiration around the nasal and perioral region could encourage polymicrobial interactions in the nasal areas and mouth [15]. Although face masks can protect against infectious microorganisms [16], especially during pandemics such as COVID-19, they might also provide a conducive environment that can breed other emerging pathogens. For example, increased sweating around the mouth caused by nose mask usage can encourage the survival of some bacteria, such as *Staphylococcus aureus* [9].

Moreover, prolonged use of face masks can unnecessarily retain heat within the nasal cavity and mouth, which might hamper some essential metabolic activities and nasal mucosal immunity, especially the nasopharynx-associated tissues, which represent the first line of immune defense [17]. The temperature alteration due to face mask usage could cause a relative increase in the dampness and warmth of the expired air, resulting in moisture condensation on the face mask user [9]. This could result in skin discomfort and encourage face-mask-associated irritation indicated by the respondents in this study. Face masks lessen the cooling impacts on facial temperatures [18], probably due to their tightness, as they cover both the nose and the mouth, leading to bad breath, breathing difficulties/resistance, and an itchy nose, which could account for the non-adherence to the use of face masks observed in this study.

Kisielinski et al. [19] indicated that unnecessary covering of some sensitive parts of the face, including the mouth and nose, with masks or protective guards elevates thermal sensations and causes severe discomfort. Moreover, high temperatures on the cheeks beneath face masks can be discomforting [20]. Due to this sheer discomfort, users tend to touch their face masks at regular intervals, which can cause cross-contamination of the mask and hands simultaneously and further spread infections [9]. This study reported that about 20% of the participants developed acne, and about 42% developed redness on the face, similar to earlier reported adverse conditions [11,21]. Using a single mask repeatedly has been shown to harbor opportunistic pathogens that contribute to acne development and sore throat [22]. Moreover, there is possibility of inhaling trace elements which are toxic, some of which are carcinogenic, with consequent nasal discomfort, oral toxicity, and irritation [23]. It has also been reported that nose mask usage worsened pre-existing rosacea, acne, and seborrheic dermatitis [21]. Frequent breaks for shorter periods of mask use can reduce the possibility of breathing difficulties and resistance [10,24]. Preventive measures such as moisturizers, emollients, and barrier creams could prevent mask-related infections [25]. However, it is essential to ensure that dressings, moisturizers, and lotions do not interfere with the mask's seal, decreasing protection against COVID-19 infection.

## 4. Conclusions

This study provides baseline data on the side effects of using face masks. This includes headaches, skin irritation, ear pains, breathing difficulties, sore throat, dry eyes, itchy nose, nasal blockage, and increased sweating around the mouth. This can discourage the widespread use of face masks and hence less adherence, which could contribute to an increased risk of infection. While COVID-19 lingers and the management of its undesirable effects persists into the future, a superior mask design must be developed with a focus on comfort, tolerability, and safety. Repeated use of a single mask should be discouraged to reduce exposure to infections. Moreover, possible precautions and standard quality control measures are necessary to ensure that masks are free from toxic substances that can endanger the health of their users.

**Supplementary Materials:** The following supporting information can be downloaded at: https://doi.org/10.7910/DVN/7QH1LY and https://doi.org/10.6084/m9.figshare.20425620, Table S1: Nose Masks and COVID, File S1: Questionnaire format.

**Author Contributions:** F.G. and A.I. conceptualized and designed this study. A.I., F.G., E.D. and M.O. prepared the survey questions. All the authors processed, interpreted, and analyzed the data. A.I. and F.G. prepared the first draft of the manuscript. A.I. and L.M. revised the draft for important intellectual content. All authors have read and agreed to the published version of the manuscript.

**Funding:** Francis Gyapong and Maame Oforiwaa were supported by funds from a World Bank African Centre of Excellence grant (ACE02-WACCBIP) and a DELTAS Africa grant (DEL-15-007) from Gordon Awandare of the West African Centre for Cell Biology of Infectious Pathogens (WACCBIP).

**Institutional Review Board Statement:** This is an epidemiological survey and not an experimental study; hence, no human or animal samples were obtained or analyzed. However, this study was approved by the Ethics Committee for Basic and Applied Sciences on 28 January 2022 (ECBAS 076/20-21) at the University of Ghana.

**Informed Consent Statement:** Not applicable.

**Data Availability Statement:** The data presented in this study are openly available in the Side Effects of Nose Masks Usage dataset at https://dataverse.harvard.edu/dataset.xhtml?persistentId=doi:10.7910/DVN/7QH1LY (accessed on 3 August 2022).

**Acknowledgments:** The authors would like to thank all the anonymous volunteers who completed the online survey and the members of the Mosi Lab and AMR Research Group (led by Abiola Isawumi), especially Molly Kukua Abban, Eunice Ampadubea Ayerakwa, and Edwin Kyei-Baffour for their technical support.

**Conflicts of Interest:** The authors declare no conflict of interest.

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
