# Peer review of "Challenges and Adverse Effects of Wearing Face Masks in the COVID-19 Era"

_challenges, doi:10.3390/challe13020067_

Round 1

Reviewer 1 Report

Thank you for the manuscript. Following are a few suggestions to improve the manuscript.

1.    The literature review is very limited in the introduction and must be improved to include more relevant studies.

2.    A paragraph is also needed at the end of the introduction highlighting the gap, objectives etc.

3.    The quality of graphs can be improved with horizontal major grid lines will be useful to read values.

4.    More statistical parameters such as interquartile range or further elaboration is needed (see Figure 1). The following seems to be insufficient 'All the assessed side effects were statistically significant with < 0.05."

5.    Please clear the manuscript of the authors’ internal comments.

6.    How and why certain discomforts were selected. Is it based on some reference?

7.    I haven’t found any Supplementary S1 information.

8.    Conclusions are very limited, please address the main findings reinforced with quantitative information.

9.    Authors can also highlight the significance of work in view of future benefits not just limited to Covid-19.

Author Response

Reviewer 1

Thank you for the comment and suggestions.

  1. The literature review is very limited in the introduction and must be improved to include more relevant studies.

The manuscript is a brief communication, therefore it focused on relevant however precise information to justify the rationale of the study design and its purpose. Also, we have included additional information to support the study.

  1. A paragraph is also needed at the end of the introduction highlighting the gap, objectives etc.

 This has been revised and highlighted.

  1. The quality of graphs can be improved with horizontal major grid lines will be useful to read values.

 This has been revised as suggested.

  1. More statistical parameters such as interquartile range or further elaboration is needed (see Figure 1). The following seems to be insufficient 'All the assessed side effects were statistically significant with < 0.05."

Upon consideration, we have removed 'All the assessed side effects were statistically significant with < 0.05." Also, the data is categorical and was presented in descriptive bar graphs with statistical justification. In our opinion, the statistical parameters addressed the data generated with appropriate relevance. We appreciate and acknowledge the reviewer’s suggestions; however, it might not add more information to as already stated.

  1. Please clear the manuscript of the authors’ internal comments.

  1. How and why certain discomforts were selected. Is it based on some reference?

As highlighted in the manuscript, selected discomforts were adopted as referenced (4, 8, 9, 10, 11). Others were as a result of information associated with use of nose masks, directly or indirectly before COVID-19 pandemic as referenced (12).

  1. I haven’t found any Supplementary S1 information.

 This has been included - https://doi.org/10.7910/DVN/7QH1LY

  1. Conclusions are very limited, please address the main findings reinforced with quantitative information.

The conclusion justified the aim of the study and highlighted the relevant information from the data generated. We have included an additional statement.

  1. Authors can also highlight the significance of work in view of future benefits not just limited to Covid-19.

We have indicated this as a part of the conclusion especially with overuse of a single mask which can increase exposure to infections. Also, ensuring the masks pass through strict quality control test.

Reviewer 2 Report

The article by Gyapong et al. fits within the aims and scope of Challenges, however this manuscript needs work before I would be willing to accept it. Currently, this manuscript needs major revisions.

The authors use the word “nose” this implies that the mask only covers the nose and not the mouth. I highly suggest the authors change the word from “nose” to “face” if these are the types of masks that they are analyzing within the manuscript. The majority of the masks used during the pandemic were face masks that covered the mouth and the nose. The usage of the word nose is a major concern of mine that I cannot ignore. If the authors did conduct the study with just “nose” masks I believe it would be appropriate to show a diagram with someone covering just their nose and not there mouth in a separate figure that way the audience can visualize this type of mask usage.

The authors state breathing difficulties in the results and discussion but I am not sure how this can be the case if the mouth is not covered.

Within figure 1 to me it looks like it says No comfortability, please put the word comfortability in the middle. The error bar for no is off the chart, this needs to be within the chart.

For figure 4(a) there needs to be consistency of where the numbers are placed within the graph either outside or inside the bar graph. Also, it looks like the words in the y ‘axis” were just copied and pasted in, you can definitely tell this with figure 4(b).

I would like to ask the authors how nose masks could unnecessarily retain heath within the nasal cavity and mouth if the mask is only covering the nose?

The authors must go into more details on why there might be irritability, sore through, nasal discomfort etc. I highly recommend that the authors read, cite and discuss the following document within this manuscript as the following article will help address these points. Chris Douvris. "Quantification of trace elements in surgical and KN95 face masks widely used during the SARS-COVID-19 pandemic." Science of The Total Environment 814 (2022): 151924.

The conclusions are weak and need to be strengthen.

The online questionnaire needs to be included, I could not find the questionnaire within the manuscript and there was no supplemental S1? If the questions involved just the word “nose” this would make the data suspect as the participants could have inferred just nose coverings.

The authors need to address my points before I would be willing to accept this manuscript. Currently this manuscript needs major revisions.

Author Response

Reviewer 2

The article by Gyapong et al. fits within the aims and scope of Challenges, however this manuscript needs work before I would be willing to accept it. Currently, this manuscript needs major revisions.

The authors use the word “nose” this implies that the mask only covers the nose and not the mouth. I highly suggest the authors change the word from “nose” to “face” if these are the types of masks that they are analyzing within the manuscript. The majority of the masks used during the pandemic were face masks that covered the mouth and the nose. The usage of the word nose is a major concern of mine that I cannot ignore. If the authors did conduct the study with just “nose” masks I believe it would be appropriate to show a diagram with someone covering just their nose and not there mouth in a separate figure that way the audience can visualize this type of mask usage.

The authors state breathing difficulties in the results and discussion but I am not sure how this can be the case if the mouth is not covered.

We defined ‘nose masks’ in the study to include ‘masks covering the nose and mouth’. ‘Nose masks’ assessed in this study are those used during the pandemic; therefore, we didn’t see any ambiguity in the use of ‘nose masks’ as for ‘face masks’ in this study. However, for clarity, we have decided to adopt ‘FACE MASKS’ as suggested by the reviewer where appropriate within the context of the manuscript. We would maintain the use of ‘nose masks’ for published studies we referenced.

Within figure 1 to me it looks like it says No comfortability, please put the word comfortability in the middle. The error bar for no is off the chart, this needs to be within the chart.

This has been corrected and revised – Figure 1

For figure 4(a) there needs to be consistency of where the numbers are placed within the graph either outside or inside the bar graph. Also, it looks like the words in the y ‘axis” were just copied and pasted in, you can definitely tell this with figure 4(b).

This has been adjusted and revised.

I would like to ask the authors how nose masks could unnecessarily retain heath within the nasal cavity and mouth if the mask is only covering the nose?

The ‘nose masks’ as indicated in the previous comment covered both nose and mouth. We have adopted the suggestion of ‘FACE MASKS’.

The authors must go into more details on why there might be irritability, sore through, nasal discomfort etc. I highly recommend that the authors read, cite and discuss the following document within this manuscript as the following article will help address these points. Chris Douvris. "Quantification of trace elements in surgical and KN95 face masks widely used during the SARS-COVID-19 pandemic." Science of The Total Environment 814 (2022): 151924.

Thank you for this recommendation. We have included this in the discussion – Reference 24

The conclusions are weak and need to be strengthen.

We think the conclusion of the study addressed the data generated with appropriate inferences. We don’t want to make general assumptions; however, we have included additional statement in the conclusion section.

The online questionnaire needs to be included, I could not find the questionnaire within the manuscript and there was no supplemental S1? If the questions involved just the word “nose” this would make the data suspect as the participants could have inferred just nose coverings.

The questionnaire was clear and properly laid-out. The respondents had proper clarity to the use of nose masks as those covering nose and mouth, with appropriate responses accordingly. For example, we have some of the staff of our institute participating in the study with no difficulties and ambiguity of ‘nose masks’ as ‘face masks’.

We have included a link to this -  https://doi.org/10.7910/DVN/7QH1LY

The authors need to address my points before I would be willing to accept this manuscript. Currently this manuscript needs major revisions.

All the points have been properly considered where appropriate; and reasonable justifications have been provided where necessary.

Round 2

Reviewer 1 Report

The comments are addressed satisfactorily. 

Reviewer 2 Report

I would like to thank the authors for there efforts for addressing my questions and concerns, the manuscript has been greatly improved since the first version. The only point I would like to make is within the reference section that the font associated with reference number 24 needs to be increased. Other than this minor correction this manuscript is ready to be accepted.